# Modeling Future Land Use Development: A Lithuanian Case

**Gintautas Mozgeris \*** and **Daiva Juknelienė**

Agriculture Academy, Vytautas Magnus University, Studentų Str. 11, LT-53361 Akademija,
44248 Kauna, Lithuania; daiva.jukneliene@vdu.lt
\* Correspondence: gintautas.mozgeris@vdu.lt

**Abstract:** Effective management decisions regarding greenhouse gas (GHG) emissions may be hampered by the lack of scientific tools for modeling future land use change. This study addresses methodological principles for land use development scenario modeling assumed for use in processes of GHG accounting and management. Associated land use policy implications in Lithuania are also discussed. Data on land uses, available from the National Forest Inventory (NFI) and collected for GHG accounting from the land use, land use change and forestry (LULUCF) sector in the country, as well as freely available geographic information, were tested as an input for modeling land use development in the country. The modeling was implemented using the TerrSet Land Change Modeler. Calibration of the modeling approach using historical land use data indicated that land use types important for GHG management in the LULUCF sector were predicted with an accuracy above 80% during a five-year period into the future, while the prediction accuracy for forest and built-up land was 96% or more. Based on several land management scenarios tested, it was predicted that the LULUCF sector in Lithuania will accumulate $CO_2$, with the forest land use type contributing most to $CO_2$ absorption. Key measures to improve the GHG balance and carbon stock changes were suggested to be the afforestation of abandoned or unused agricultural land and prevention of the conversion of grassland into producing land.

**Keywords:** land use; land use change; scenario; carbon stock changes; simulation; forest; producing land; grassland



## 1. Introduction

Substances of anthropogenic origin have a major influence on the climate system [1]. Human economic activity contributes to thermal atmospheric pollution—increasing greenhouse gas (GHG) concentration enlarges the natural greenhouse effect and plays a decisive role in the rise of the average global temperature [2–4]. GHGs are mainly generated by the combustion of fossil fuels in industrial and agricultural production processes, and, by a large proportion, from waste [3,5–9]. GHG absorption is usually associated with the physiological properties of green vegetation, as other solutions to sequester carbon have not yet been proven to be either technologically or economically effective [10,11]. Meanwhile, climate change is a global issue and needs to be addressed through global cooperation among countries to improve energy efficiency, develop and deploy clean technologies, and increase natural GHG absorption. In this context, the processes in and around land use, land use change, and forestry (LULUCF) are becoming crucially important. The LULUCF sector includes GHG emission and its removal from forests, arable or producing land, grasslands and pastures, wetlands, built-up areas, and other land plots. Emissions and removals of GHGs are accounted using internationally accepted approaches [12–14]. However, in order to actively increase carbon absorption, it is necessary to know and manage the processes involved in the development of land surface layers and land use. Cognitive processes and management decisions will be hampered by a lack of access to scientifically based tools for modeling land use and hence GHG emissions.

Nowadays, many land use change modeling tools exist, differing in their methodological implementation [15,16]. They may cover universal or very specific application fields, with the focus on local case studies or continental exercises. There are several concepts of land cover and use modeling [17]—economic models, system dynamics approaches, cellular automata, and agent-based models. Spatial economic or econometric models deliver generalized predictions of states of phenomenon by balancing various inter-related input factors determining their development. System dynamics or causality-driven models assume an empirical modeling of land cover or land use changes. This involves (i) an assessment of past changes first, (ii) a determination of relationships between land changes and factors driving such changes, (iii) an evaluation of change potential, and (iv) an allocation of land to the new land cover or land use types [18]. Cellular automata usually operate in a raster domain, representing the landscape as an n-dimensional grid of cells. Each cell may acquire a finite number of states, which may change over time following some set of rules and depending on the state of neighboring cells. Models are iterated over time, delivering land cover or land use status within the cell at specific times [19,20]. Agent-based models are aimed at modeling the behavior of autonomous individuals (agents) who may perceive their environment and interact with individuals [21]. Even though there are numerous potential solutions for land use change modeling, their applicability is heavily restricted by various legal, technological, and organizational aspects. The land use change modeling depends on the specific requirements of GHG emission accounting, the availability and specifics of input data, modeling tools, and experiences, especially when considering specific countrywide exercises.

There are many factors influencing GHG emissions and absorptions in the LULUCF sector, potentially resulting in uncertainties in both GHG accounting and projections [22–26]. Simultaneously, availability, or often the lack of input data for land use change analysis, makes the task more challenging [27]. Even though there are international standards to account for GHGs, there are always some specialties present in the operational approaches of each country. Lithuania, following its international climate change mitigation commitments, has developed an original LULUCF monitoring system, which is used for GHG reporting. This system predetermines the approaches of land use development projections. The core data source for GHG accounting from the LULUCF sector in the country is the National Forest Inventory (NFI), which is implemented by the State Forest Service [28,29]. Originally developed to provide statistical information on forest resources for strategic forestry planning at a country level, the Lithuanian NFI has recently been expanded to collect countrywide data on land uses and land use changes. The land uses are monitored in a systematic network of observation points through the whole country, while forest attributes are surveyed at points in the forest. There are operational solutions introduced in Lithuania to model the development of forest resources and forestry, ranging from forest stand-level simulators to systems manipulating aggregated countrywide data [30–32]. The State Forest Service uses the European Forestry Dynamics Model (EFDM), developed as a harmonized forestry modeling tool for all European countries, based on NFI data. The EFDM has been used to calculate the forest reference level (FRL) for Lithuania following the European Union LULUCF regulation for 2021–2030 [13]. The EFDM is a matrix-based model of a Markov chain type representing change by transition of areas (in this case, the NFI sample plots) between different fixed states of the forest [33]. This matches well with the system dynamics or causality-driven models introduced above. The reference levels for land uses other than forest are based on historical data, thus, one may assume that no sophisticated modeling solution is needed. Nevertheless, successful land use management provides challenges for modern decision-support tools which are based on land use development scenarios. To our knowledge, the solution that has been widely used to make GHG projections in the LULUCF sector in Lithuania has been the land use, land use change and forestry emission accounting tool, LULUCFeat [34]. LULUCFeat delivers GHG predictions based on aggregated LULUCF data and past trends, using information on driving factors and expert knowledge. Methodologically, this fits the economic models mentioned above.

However, the solution is too focused on delivering certain GHG reports and underfitting expectations for a versatile land use change modeling system, based on all NFI data and compatible modeling principles.

Thus, the aim of the study introduced in this paper is to test the methodological principles for land use development scenario modeling for use in processes of GHG accounting and management. First, we ask what is the performance of the Markov chain analyses methodological approach in modeling land use development using standard GIS software? To conduct the modeling exercise, we use inputs available from already running in Lithuania inventory projects and freely available geographic databases. Then, we test the capacity of the LULUCF sector in Lithuania to accumulate carbon during the next decade, starting in 2020. For that, we project the development of major land use types in Lithuania until 2030 using several land use management scenarios and estimate potential contributions of different land uses on carbon emission/absorption. We hypothesize that the carbon accumulation in the LULUCF sector in Lithuania during the next decade should increase. Finally, we end with a discussion and proposals for both methodological enhancements of modeling solutions and land use management policies.

## 2. Materials and Methods

### 2.1. Study Area

The study was conducted in Lithuania, located in Central Europe (Figure 1) and having historically strong links with Eastern Europe. Land use development in Lithuania in recent decades strongly depended on the radical societal transformations after Lithuania broke away from the Soviet Union in 1990 and later joined the European Union in 2004 [35]. The area of three land uses important in GHG accounting and management (forest, producing land and grassland) was rather similar (around 28–30%) in 1971. Then, the proportions of forest, wetland, built-up areas, and other land use types changed relatively steadily since 1971, while the trends of producing land and grassland development changed their trajectories around 1990 and again about 2005 (Figure 2). The proportions of forest land and producing land in 2015 were, respectively, 34% and 33%. The proportion of grassland was reduced to 23%, and the proportions of both wetland and built-up land were 5%. It should be noted that the total area of Lithuania is 65,200 km$^2$.

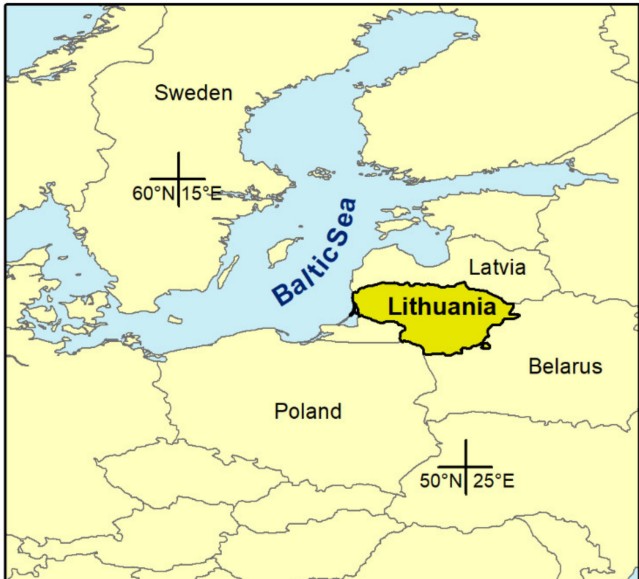

**Figure 1.** Location of the study area. Source of the data used: https://thematicmapping.org/downloads/world_borders.php (accessed on 22 March 2021).

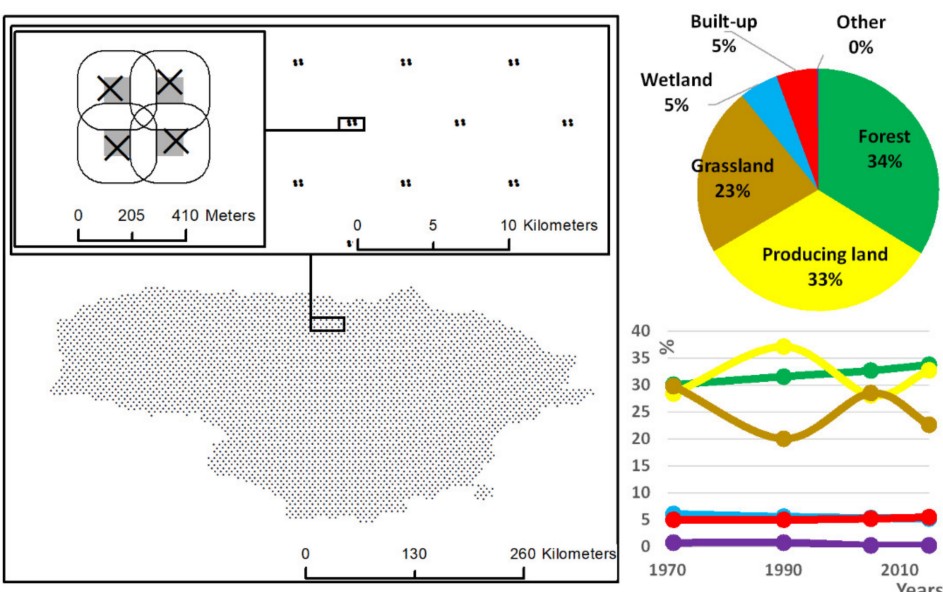

**Figure 2.** Specification of the study area: (**left**) associative illustration of the distribution of sample plots at different scales, with gray squares referring to the 100 × 100 m cells associated with Lithuanian National Forest Inventory (NFI) sample plots with 100 buffers used to extract driver variables for land use change modeling; (**right**) proportions of major land use types in Lithuania in 2015 and changes in proportions since 1971. Source of the data used: Lithuanian National Forest Inventory.

### 2.2. Input Data

Two types of input data were used in the study: (i) data describing the land uses in Lithuania and (ii) data describing the factors influencing the land use changes. Land use information was available from the Lithuanian NFI [29,36]. Land use types and subtypes have been identified annually on a network of 16,349 systematically distributed sampling points (Figure 2) since 1971 using the nomenclature of GHG inventories [37]. Usually, three levels of identification are used; however, we used only two levels in our study, i.e., Level 1 with 6 land use types (forest land, producing land, grassland and pastures, wetland, built-up areas, and other land) and Level 2 with 25 subtypes specifying the types in more detail (Appendix A, Table A1 provides a full list of land use subtypes). To conduct the modeling and to integrate the NFI data with other datasets, we created a raster map with a cell size of 100 × 100 m and assured that each NFI plot was associated with a unique cell. Only cells with an NFI plot were used for the study. Free data available from the spatial information portal of Lithuania (www.geoportal.lt, accessed on 22 March 2021) were used to describe the factors influencing the land use changes. The following geographic datasets were used: GRPK (spatial dataset of (geo) reference base cadaster), GDR50LT (georeferenced spatial dataset for the territory of the Republic of Lithuania at the scale of 1:50,000), AZ_DRLT (spatial dataset of abandoned land of the territory of the Republic of Lithuania), SŽNS_DR10LT (database of limited land use areas of the Republic of Lithuania at scale 1:10,000), Dirv_DR10LT (spatial dataset of soil of the territory of the Republic of Lithuania at scale 1:10,000), KŽS (land parcel identification system database), the spatial dataset on the farmland, cropland, and crop types from the National Paying Agency under the Ministry of Agriculture and Population, and the 2011 housing census data from Lithuanian official statistics portal (https://osp.stat.gov.lt/documents/10180/1491916/WHOLE.zip, accessed on 22 March 2021). Two approaches were used to specify the explanatory variables: (i) the area of specific features within a 100 m buffer zone around each 100 × 100 m cell associated with the NFI sample plot was estimated, and (ii) the shortest distance from the NFI sample plot center to specific features was estimated. All explanatory variables were stored as raster maps with a cell size of 100 × 100 m. Optimization of the explanatory variables is described in the next subchapter.

### 2.3. Modeling Land Use Development

Modeling of the land use development was implemented using the TerrSet 18.21 Land Change Modeler [38]; thus, some approaches used were predefined by the functionality of the available tools. Therefore, the modeling started with an analysis of land use changes between two points in time. The potential of land use transitions was then modeled using a set of driver or explanatory variables. A set of maps of suitability for each transition was developed. Based on land use changes in the past, probabilities of land use change in the future were calculated by building a matrix with probabilities of all possible land use changes. Finally, the land use changes were predicted using the historical rates of change and the transition potential models for a specified date in the future.

Our study consisted of two stages. First, we calibrated and validated land use change modeling using input data freely available in Lithuania. We then simulated land use development for the next decade using several land use change scenarios. The methodological framework of our study is summarized in Figure 3.

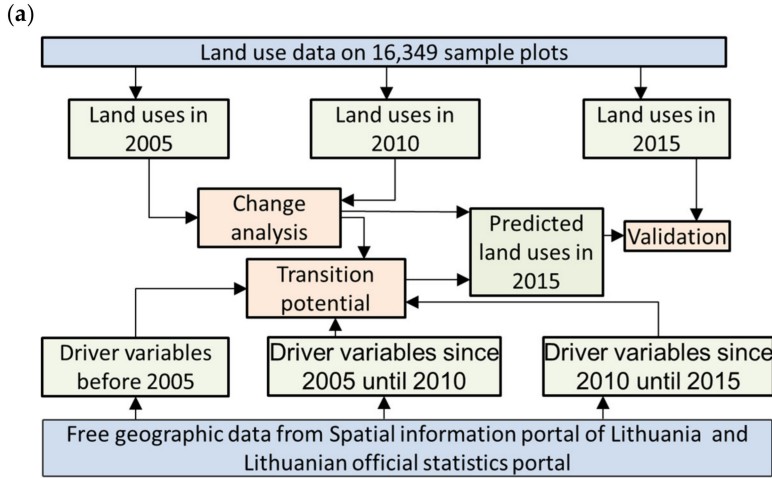

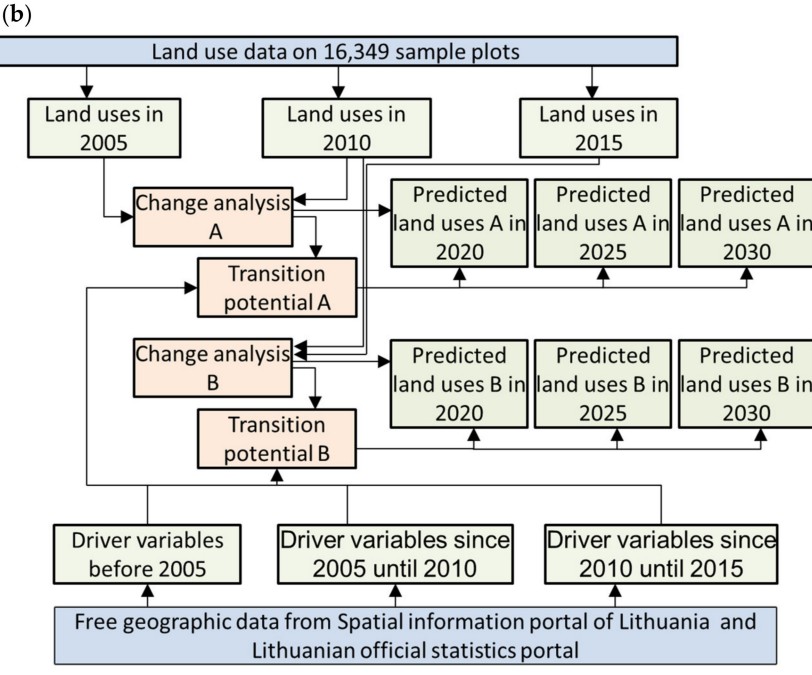

**Figure 3.** Flowchart summarizing the overall structure of the study: (**a**) calibrating and validating the land use change model, and (**b**) modeling land use development until 2030.

We first analyzed the land use development during the period from 2005 to 2010 to predict land uses in 2015. Transitions were modeled using a multilayer perceptron (MLP) neural network algorithm. All driver variables were tested before using them to build the transition potential models. First, Cramer's V statistic was calculated for each potential explanatory variable—only variables that had a Cramer's V of 0.15 or higher were considered as having potential for modeling. The variable with a higher Cramer's V statistic was considered for modeling among highly intercorrelated variables. Finally, the lists of driver variables were optimized, analyzing the modeling reports delivered by the TerrSet system and iterating the final lists of variables that produced the best MLP performance. All driver variables were considered static. Six strategies were tested to include the driver variables in building the transition potential models, differing by the number and type of driver variables, the date they referred to, and the preprocessing solutions (Table 1).

**Table 1.** Tested strategies for the inclusion of driver variables in building the transition potential models (+ means that the variables from the respective group were considered or an optimization of variables was applied).

| Strategy of Using Driver Variables | Versions of KŽS | | | AZ_DRLT, SŽNS_DR10LT, Dirv_DR10LT, and Census Data | Land Use Declaration Data | Optimization of Explanatory Variables |
|:---:|:---:|:---:|:---:|:---:|:---:|:---:|
| | Before 2005 | Between 2005 and 2010 | After 2010 | | | |
| 1 | | + | | + | | |
| 2 | + | + | | + | | |
| 3 | | + | + | + | | |
| 4 | + | + | + | + | | |
| 5 | + | + | + | + | + | |
| 6 | + | + | + | + | + | + |

Driver variables originating from the KŽS database were grouped according to the date they were created: variables based on data collected (i) before 2005, (ii) between 2005 and 2010, and (iii) after 2010. This was aimed to simulate exercises, where variables changing over time were considered land use development scenario specifications. For example, variables collected after 2010 did not influence the land use change before 2010, but they could be used to specify the future (actual or expected) dynamics of factors influencing land uses. The land use declaration data from the spatial dataset on farmland, cropland, and crop types refer to 2012. The most current versions of other datasets were used. A full list of explanatory variables considered is provided in Appendix A, Table A2. Future land use was predicted using a hard prediction model. The quantity of change in each transition was modeled through a Markov chain analysis.

The second stage of our study included predicting land use development in the future, i.e., acquiring the areas of major land use types for 2020, 2025, and 2030. The sixth strategy using driver variables was applied, i.e., all available explanatory variables were tested before use in the transition potential models. Two options of actual land use change were considered to build the Markov matrix, i.e., (i) the changes from 2005 to 2010 and (ii) from 2010 to 2015. The land use change scenarios were also specified by editing the Markov matrix. The land use development scenarios considered are introduced in Table 2.

**Table 2.** Description of future scenarios of land use change.

| Scenario Title | Main Features for Building the Markov Matrix | |
| --- | --- | --- |
| | Period | Manual Transformations of Transition Probabilities |
| Reference (2005–2010)<br>Reference (2010–2015) | 2005–2010<br>2010–2015 | - |
| Producing land to forest (2005–2010) | 2005–2010 | The probability of transformation of the following land into the forest is doubled: arable land, natural grassland with trees and brush, brush |
| Producing land to forest (2010–2015) | 2010–2015 | The probability of transformation of arable land into cultural grassland and pastures is doubled, and the remaining natural grassland with trees and brush is transformed into cultural grassland and pastures |
| Grassland to forest (2005–2010)<br>Grassland to forest (2010–2015) | 2005–2010<br>2010–2015 | All natural grasslands with trees and shrubs are transformed into forest land. |
| No grassland to producing land (2005–2010) | 2005–2010 | There is no transformation of grassland/pasture land into producing land, and all other land use changes follow trends during the reference period |
| No grassland to producing land (2010–2015) | 2010–2015 | |

To obtain approximate indications of potential contributions of different land uses on carbon emission/absorption, we applied average conversion factors for 2015, as used to prepare the national GHG report from the LULUCF sector [39]; i.e., the following emission values in tons of $CO_2$ equivalent per ha were used: forest land, 3.93; producing land, 1.43; grassland, 0.51; wetland, 2.64; and built-up land, 1.6; other land, 6.25.

*2.4. Validation Approaches*

Approaches originating from remote sensing were used to validate the performance of land use prediction. Land use types for the year 2015 were predicted on all NFI sample plots, and the predictions were compared with actual land use types recorded by the Lithuanian NFI. Error matrices were constructed where the true and predicted land use types were cross-tabulated. The validation statistics used to evaluate the prediction were the overall accuracy of prediction and Cohen's kappa:

$$Kappa = \frac{Observed\ accuracy - Expected\ accuracy}{1 - Expected\ accuracy} \tag{1}$$

$$Observed\ accuracy = Overal\ accuracy = \frac{tp}{N} \tag{2}$$

$$Expected\ accuracy = \sum_{i=1}^{k} \frac{nt_i}{N} \times \frac{nc_i}{N}, \tag{3}$$

where *tp* refers to the number of samples predicted to be positive that are, in fact, positive, *k* refers to the number of classes, $nt_i$ refers to the number of samples truly in class *i*, $nc_i$ refers to the number of samples assigned to class *i*, and *N* refers to the total number of samples.

The interpretation of Cohen's kappa was as follows: under 0: "poor"; 0–0.2: "slight"; 0.2–0.4: "fair"; 0.4–0.6: "moderate"; 0.6–0.8: "substantial"; 0.8–1.0: "almost perfect" [40].

Land use type–specific prediction performance was evaluated using precision (producer's accuracy), recall (user's accuracy), and the *F*-score (the harmonic mean of recall and precision):

$$Precision = \frac{tp}{tp + fp} \tag{4}$$

where *fp* refers to the number of samples predicted positive that are, in fact, negative;

$$Recall = \frac{tp}{tp + fn} \tag{5}$$

where *fn* refers to the number of samples predicted negative that are, in fact, positive;

$$F\text{-score} = 2 \times \frac{Recal \times Precision}{Recal + Precision} \tag{6}$$

The *Z* statistic was used to test whether two prediction error matrices were statistically different:

$$Z = \frac{|\hat{\kappa}_1 - \hat{\kappa}_2|}{\sqrt{var(\hat{\kappa}_1) + var(\hat{\kappa}_2)}}, \tag{7}$$

where $\hat{\kappa}_1$ and $\hat{\kappa}_2$ are the Cohen's kappas of compared predictions, and $var(\hat{\kappa}_1)$ and $var(\hat{\kappa}_2)$ refer to the variances of the respective matrices. Compared predictions were treated as statistically differing if *Z* was more than 1.96 [41].

## 3. Results

### 3.1. Calibration and Validation of Land Use Change Models

First, we predicted all land uses in 2015 for all sample points using the input data for 2005–2010 and assuming the Reference scenario. The overall accuracy of prediction was in the range 82–83% (Table 3). The kappa statistic was 0.76–0.77. It seems that the factor inclusion strategy in the calculation of transformation potential had no significant effect on prediction accuracy; the kappa statistics did not differ with statistical significance, with the highest value of the *Z* statistic being 0.148 (not presented in Table 3). The prediction accuracy statistics of the most encountered land use classes are summarized in Figure 4. The most accurately predicted land cover class is forest land—both the producer's and user's accuracies yielding nearly 99%. The development of built-up areas is also accurately predicted; the F-score is 96%. It is noteworthy that practically in all cases the producer's accuracy (~94.5%) is lower than the user's accuracy (~97.5%), suggesting that other land use classes are more often incorrectly predicted to be transformed into built-up land, rather than vice versa.

The accuracy of predicting the producing land was notably better than that of cultural grassland/pastures, natural grassland, or natural grassland with trees and brush. On average, producing land was predicted with 84–87% accuracies, and the producer's accuracy was higher than the user's accuracy. Cultural grasslands/pastures, natural grasslands, and natural grassland with trees and brush resulted in the lowest prediction accuracies (if considering the most abundant land uses). Only the prediction accuracy for cultural grasslands/pastures reached 50%, and the producer's and user's accuracies did not differ. More in-depth analysis of error matrices confirmed that the abovementioned land uses were mixed with each other during the prediction. Therefore, cultural grasslands/pastures, natural grassland, and natural grassland with trees and brush are combined into one class—grassland. Following this combination, the overall classification accuracy increased by 7–8%, but the increase in kappa was not statistically significant (Table 3). After the merge, grasslands were predicted with 73–80% accuracy, and the producer's accuracy was lower than the user's accuracy. Land with brush was predicted with ~60% accuracy, but the area of this type was relatively small.

The modeling exercise was repeated using the assumptions of Scenario 3: no grassland to producing land (2005–2010). Although the overall prediction accuracy improved by 1–2%, this improvement is not statistically significant. Different scenario conditions had an impact in predicting the grassland development when using detailed grassland subtypes. After combining the grassland subtypes, we achieved very similar producer's and user's accuracies, i.e., differing by no more than 1%.

### 3.2. Land Use Changes in the Future

Predicted proportions of three major land use types—forest land, producing land, and grassland—are presented in Figure 5. The proportion of forest land is expected to increase regardless of the scenario. It should be noted that scenarios involving active efforts to increase the area of forest land result in larger forest land areas, although never exceeding 37%. Using the land use trends from 2010–2015 to model the transition potential resulted in larger forest land proportions. The area of producing land is expected to increase only if using 2005–2010 land use data to model the transition potential. However, if extrapolating the trends from 2010–2015, the areas of producing land decrease. Manual adjustment of Markov matrices, aimed to specify additional land use policy measures, resulted in even fewer areas of producing land, if compared with the Reference scenarios. If the land use changes during 2010–2015 continue into the future, the proportion of producing land in Lithuania will be reduced to below 30%. The area proportion of grassland is increased if considering the trends during 2010–2015 and, vice versa, decreased if using the 2005–2010 period to model the transition potential. The exception was the scenario with no grassland for producing land, where the grassland decrease stopped by adjusting the Markov matrix. If the land use change trends during 2010–2015 continue in the near future, the proportion of grassland will be projected to increase to 23–28%, depending on the scenario. The lowest grassland proportions were achieved in the scenario of grassland to forest (2005–2010), i.e., following the fast decreasing grassland areas from the half decade, since Lithuania joined the EU and introduced measures for grassland conversion into forest land. It should be noted that the projected trends of producing land development are inversely followed by the trends of grassland proportion.

**Table 3.** Prediction accuracy of all tested land use types.

| Strategy of Using Driver Variables | All Land Use Subtypes | | Grasslands Merged into One Class | | Z Statistics |
|---|---|---|---|---|---|
| | Overall Prediction Accuracy | Kappa | Overall Prediction Accuracy | Kappa | |
| Scenario: Reference | | | | | |
| 1 | 81.9 | 0.76 | 87.7 | 0.83 | 1.296 * |
| 2 | 82.1 | 0.76 | 88.0 | 0.84 | 1.310 * |
| 3 | 82.2 | 0.76 | 88.3 | 0.84 | 1.361 * |
| 4 | 82.1 | 0.76 | 88.2 | 0.84 | 1.369 * |
| 5 | 82.8 | 0.77 | 88.6 | 0.84 | 1.295 * |
| 6 | 81.9 | 0.76 | 88.9 | 0.86 | 1.783 * |
| Scenario: No grassland to producing land (2005–2010) | | | | | |
| 1 | 82.8 | 0.77 | 89.5 | 0.86 | 0.268/0.467 ** |
| 2 | 82.9 | 0.77 | 89.6 | 0.86 | 0.228/0.414 ** |
| 3 | 83.0 | 0.78 | 89.8 | 0.86 | 0.248/0.394 ** |
| 4 | 82.8 | 0.77 | 89.5 | 0.86 | 0.235/0.349 ** |
| 5 | 83.1 | 0.78 | 89.7 | 0.86 | 0.112/0.292 ** |
| 6 | 83.1 | 0.78 | 89.8 | 0.86 | 0.254/−0.022 ** |

* all classes vs. grassland in the one-class Reference scenario, ** Reference scenario vs. scenario with no grassland for producing land (2005–2010) (in the numerator—all land use subtypes; in the denominator—grasslands merged into one class).

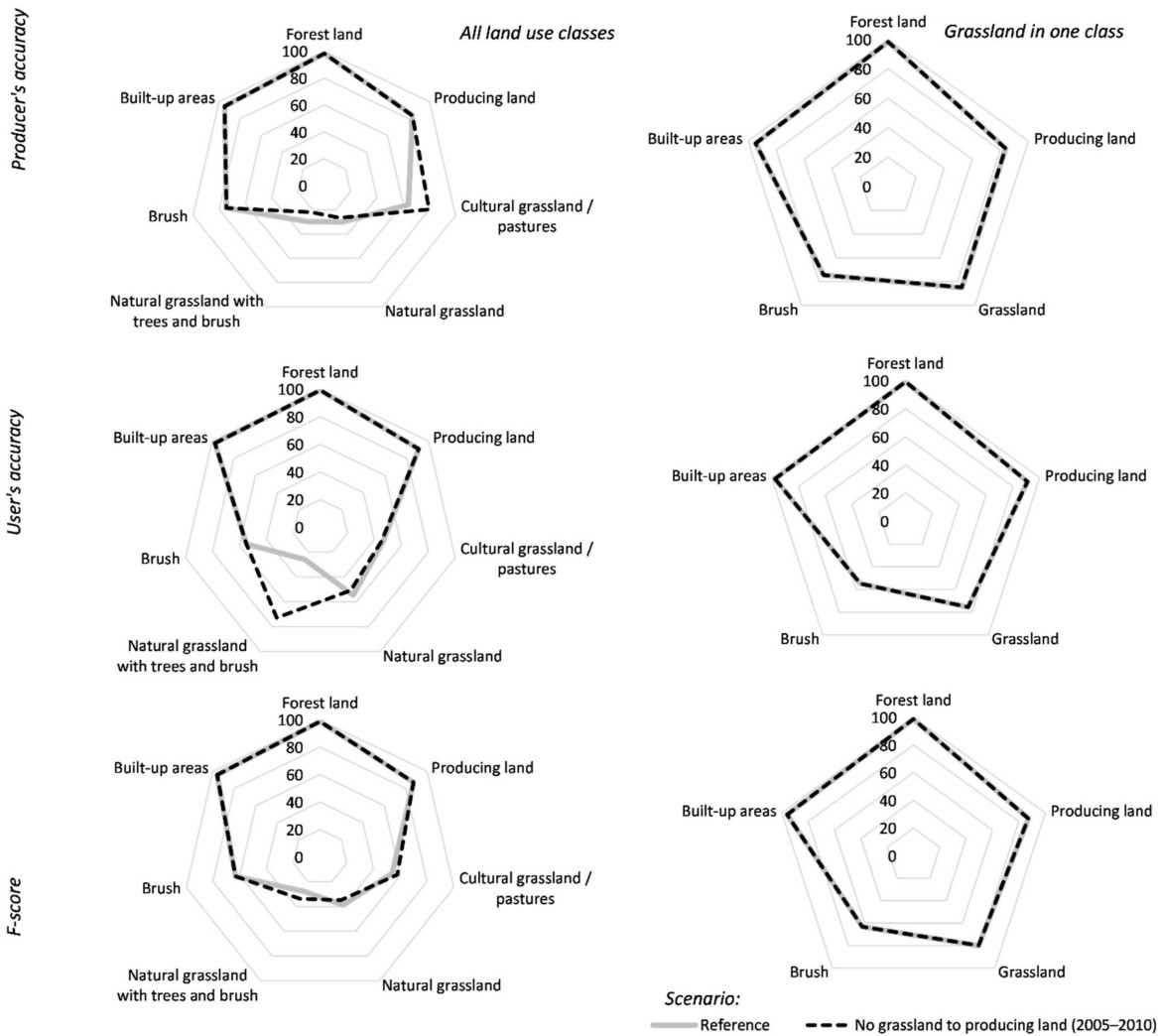

**Figure 4.** Predicting accuracy of some of the most encountered land uses, achieved using a strategy of driver variable selection based on optimization.

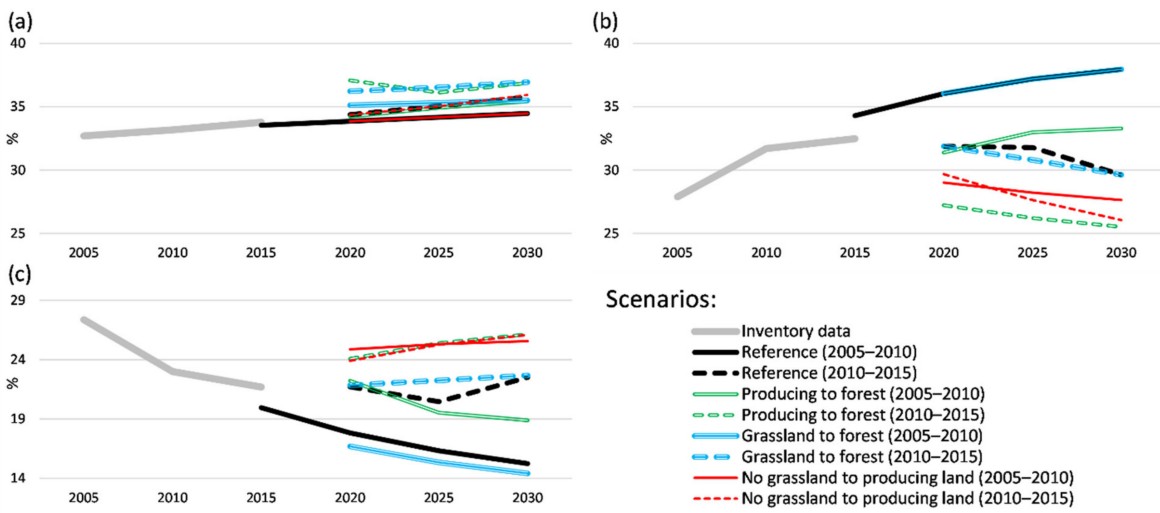

**Figure 5.** Projected development of selected land uses in Lithuania, depending on land use change scenarios: (**a**) forest land, (**b**) producing land, and (**c**) grassland.

None of the tested scenarios suggested carbon emissions from the LULUCF sector in Lithuania before 2030 (Figure 6). A larger absorption (up to 33%) was projected when considering land use changes that took place from 2010 to 2015 in modeling the transition potential. The largest overall absorption (above 1 ton of $CO_2$ equivalent from 1 ha) was achieved in the scenario where producing land became forest (2010–2015), i.e., aiming to maximize producing land conversion into forest land.

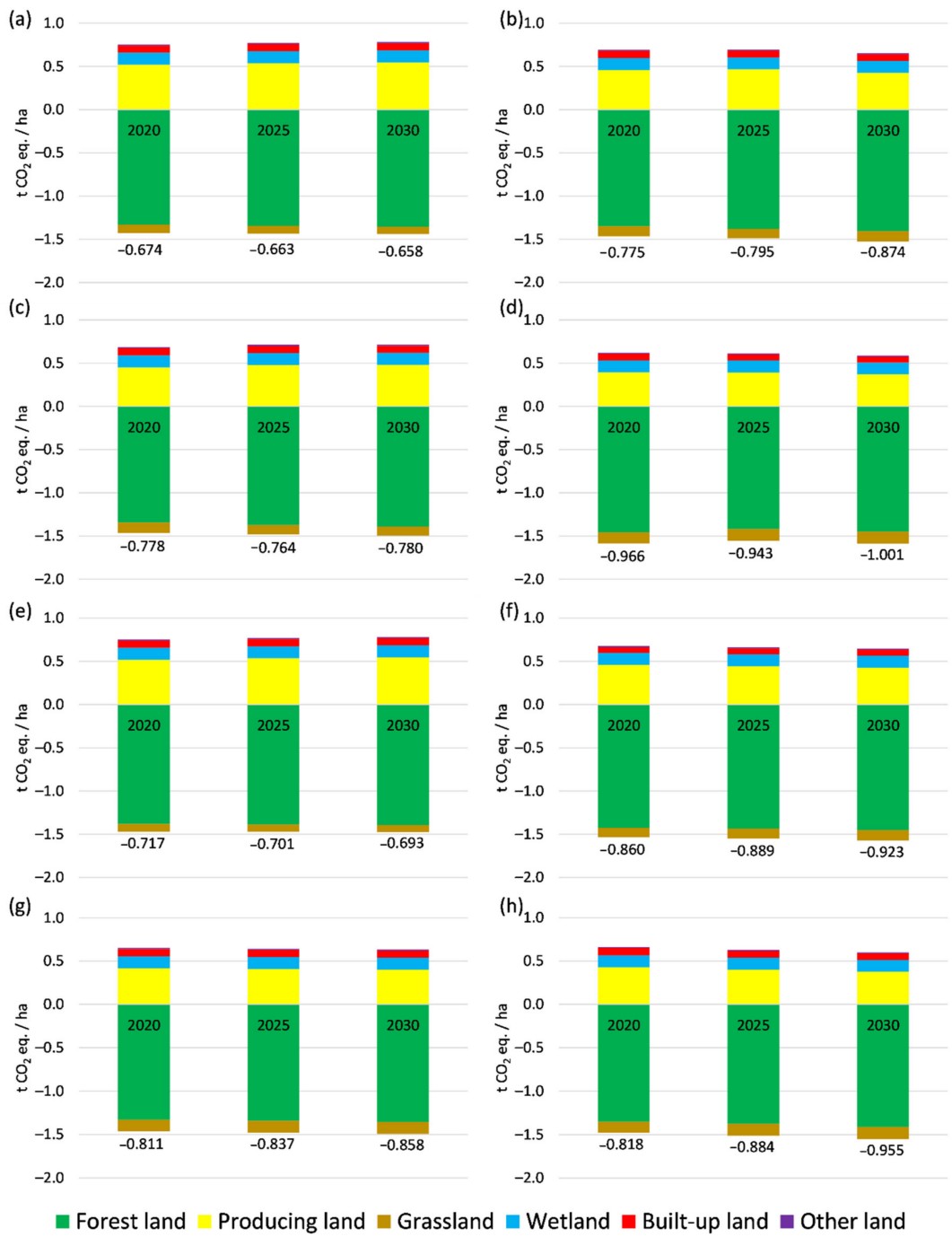

**Figure 6.** Predicted carbon emission and absorption from the land use, land use change and forestry (LULUCF) sector in Lithuania, depending on scenario: (**a**) Reference (2005–2010), (**b**) Reference (2010–2015), (**c**) producing land to forest (2005–2010), (**d**) producing land to forest (2010–2015), (**e**) grassland to forest (2005–2010), (**f**) grassland to forest (2010–2015), (**g**) no grassland to producing land (2005–2010), and (**h**) no grassland to producing land (2010–2015). The value shown below each bar indicates the total carbon sequestration value. Numeric values can be found in Appendix A, Table A3.

## 4. Discussion

There are two parts in the discussion that follows. First, we briefly address the choices, findings, and limitations related to the methodological approaches used to predict the land use changes. Second, we use our predictions to discuss potential land use development in Lithuania and the related land use policy implications.

The choice of methodological land use modeling approach was influenced both by the specificity of the general modeling environment and scientific considerations. First, the focus has been on the needs for current land use change modeling in Lithuania. We associate our study with the needs related first to the management of GHG emissions and absorption in the LULUCF sector. Thus, the input data on land use was based on information collected on a network of systematically distributed samples, inheriting differing estimation accuracies for specific land use types. Similar modeling studies usually focus on wall-to-wall land covers or land uses, even though they may be of rather coarse resolution, such as the CORINE database [27]. On the other hand, pointwise land use estimates may make the availability of information driver variables easier, as we do not necessarily need to map wall-to-wall all the factors influencing land use development. Moreover, many driver variables used in the current study are extracted using distance-based, focal, or zonal GIS analysis. Our approach is to use only publicly available information on driver variables, usually downloadable from the spatial information portal of Lithuania (www.geoportal.lt) or freely available from authorized institutions based on license agreements for research and education use. Unfortunately, we could not acquire data on land ownership, which use is commercialized using legal regulations. Finally, our methodological approach had to be compatible with that used by Lithuanian authorities to implement their international commitments, including the European Union land use, land use change, and forestry regulation for 2021–2030 [13], that is, we choose a modeling engine that is compatible with the EFDM, which has been used to calculate the forest reference level for Lithuania and already used to facilitate forest policy building processes. Last but not least, the exercise was implemented using standard GIS software packages, including both data engineering and modeling, i.e., not requiring extra efforts for software development.

The prediction of land cover or land use development in general, and the use of models of the Markov chain type in particular, has always been a very challenging exercise. The most difficult task is to evaluate the transition potential from one land use type into all possible other types. Numerous methodological approaches are reported, such as the weight of evidence [42], empirical probabilities [43], logistic regression [44], and neural network modeling [38,45]. Usually, the results achieved using different solutions are rather different, since studies address very different land use change tasks. Nevertheless, priorities are given to the use of the MLP algorithm, which is a type of neural network [18]. This was also used in our study. Two other methods implemented in TerrSet LCM, a similarity weighted instance-based machine learning algorithm and logistic regression, were rejected in the initial stages of our study, mainly due to the ability to model only one transition at a time and because additional complexity in the modeling exercise did not increase prediction accuracy.

The best prediction accuracies were achieved for land use types that followed linear development trends in recent decades, i.e., forest and built-up lands. Forest land change trends were most stable not only during the modeled period but also throughout the entire accounting period. Afforestation/deforestation is a relatively slow process in Lithuania [46], strictly regulated from a legal point of view, and therefore potentially the easiest to predict. Similarly, the development of built-up areas has also been steadily increasing since 1970. The areas of producing land were increasing in Lithuania since the country joined the EU, usually at the expense of grassland. Thus, the prediction of producing land and grassland changes is very important in supporting land use policies, especially for GHG management, because producing land is associated with carbon emissions, whereas grassland, in contrast, contributes to the carbon accumulation on average [47]. Usually, the misclassifications of producing land as grassland and vice versa were the main types of prediction errors,

e.g., ~16% of producing land points were wrongly predicted as grasslands and ~12% of grassland points were wrongly predicted as producing land. Identification of grassland management intensity or differentiating between, e.g., cultural and natural grassland, has always been a challenging task [48]. Using nomenclature for grassland subtypes that is too detailed has resulted in lower prediction accuracies because the grassland types are mixed with each other. Land use type in the Lithuanian inventory system refers to the center point of the sample plot, so the presence of single trees or brush may also be neglected during the inventory, unless the land has not yet been converted into forest land. The increase of forest land is usually very strictly controlled during the inventory, which has always been first focused on evaluating forest resources and involves precise measurements of individual trees on 500 $m^2$ circular plots [36]. Thus, we continued without specific grassland subtypes. No significant differences were found among the results obtained using different six-factor inclusion strategies for modeling transformation potentials. We explain this by the performance of the MLP, which is a type of machine learning algorithm. However, the number of input driver factors is limited in TerrSet LCM. Therefore, it is suggested that, in the future, the maximum amount of supporting information is collected and used in modeling the selected driver factors that are most strongly related to the land use transformations.

All scenarios tested suggested that the LULUCF sector in Lithuania will accumulate carbon, basically due to carbon accumulation in the forest land. Thus, a further increase of forest land area is extremely important to further contribute to GHG absorption. Nevertheless, none of the scenarios resulted in a forest land area percentage in the country exceeding 37% in 2030. According to official forestry statistics, forest land covered 33.7% of the country's area in 2019 [49]. Our prediction introduces some questions for official forest and land use policies in the country. The political objective is set to increase the forest land area in Lithuania by year 2030 to 23,000 $km^2$, i.e., 35% of the country's area [50]. Increasing the forest land area proportion in Lithuania is also among the key objectives of national forest policy, primarily associated with the management of GHG emission/absorption [51]. Abandoned or unsuitable agriculture lands are usually identified as afforestation targets in regulations for afforestation and reforestation [52]. In parallel, deforestation is strictly controlled and legally possible only upon the compensation of expenses for establishing new forest land [53]. Therefore, our simulations confirmed that the political afforestation targets can be achieved by 2030. There are no scenarios that suggested forest land reduction, yielding steadily increasing GHG absorption potential. Even though the GHG accumulation in forest land is increased most by introducing active measures to facilitate producing land or grassland transformation into forest land, the first tested option (producing land to forest scenario) most improves the GHG balance from the LULUCF sector.

Assuming that there are limited possibilities to further increase the areas of forest land or reduce built-up areas and wetlands, the key factor to improve the GHG balance in the LULUCF sector will be the proportion of producing land and grassland. If land use management as it was in the period between 2005 and 2010 continues without additional measures to support specific land use transformation types (Reference 2005–2010 scenario), the GHG emissions from agricultural land could increase over the next decade from 2020 by ~9.5%. However, if continuing land use management policies as they were after 2010 (Reference 2010–2015 scenario), GHG emissions could decrease by 20–35% compared with the Reference 2005–2010 scenario, and by 2030 the emissions from agricultural land could be reduced by 11%. Therefore, we can assume that different suggested development trends are affected by changes in Lithuanian land management. Historically, several periods have shaped Lithuanian land management in the last three decades. The largest impact on the use intensity of agricultural lands could be associated with the agrarian reform in the country, which started in 1991. This reform resulted in fully changed formats of agriculture, land management, and land use relationships, and production capacities of agricultural subjects. The second group of impacts is associated with Lithuania joining the European Union in 2004 and the availability of EU and national budget resources to

support agriculture and rural development. The factors influencing land use development are usually interdependent, and the outcomes of their inter-relationships during specific periods of socioeconomic development are shaped mainly by political and social factors, with natural conditions playing only a secondary role. Bearing in mind the time periods used to build the land use change models in our study, 2005–2010 could have been influenced by agriculture restructuring (which started in 1991–1992), and the next period is likely associated with the impacts of joining the EU and the state support for agriculture and rural development.

Additional measures to support specific land use transformation types were introduced in the models by manual adjustment of the Markov matrices. Three types of such measures are discussed. First, forcing the transformation of producing land into forest was associated primarily with the strategic forest policy objective to increase the proportion of forest land area, coupled with current land use management efforts to sustain grassland areas or at least to prevent their transformation back to producing land. The second type of measure (grassland to forest) was aimed to increase the forest land area on current grassland. It matches the first scenario, however, with no conditions regarding the efforts to prevent transformations from producing land into grassland. The third type of measure (no grassland to producing land) was associated with additional efforts to prevent grassland transformation into producing land only, i.e., leaving out the extra efforts to increase forest land area. Therefore, if a land use management policy generally follows that in effect from 2005 to 2010, additional measures to support specific land use transformation types will not result in reducing GHG emissions, either from agricultural land or the entire LULUCF sector in the decade starting at 2020. Conversely, GHG emissions from agricultural land are predicted to be reduced in the coming decade if the land use management policy used from 2010 to 2015 is followed. Introducing extra measures would support the reduction of GHG emissions from agricultural land. Especially important in this context is the reduction of producing land by its transformation into forest land (producing land to forest) or preventing the transformation of grassland into producing land (no grassland to producing land). The introduction of such measures may reduce GHG emissions in the next decade by ~16 and 28%, respectively.

Summarizing, in order to improve the GHG balance in the LULUCF sector in Lithuania over the next decade starting at 2020, the focus in Lithuania should be to increase forest and grassland areas. This objective is supported by national strategic political documents, especially those aimed at the effective use of EU support [54–56]. The key contributor to the total $CO_2$ balance in the LULUCF sector will remain the total forest land area and the potential to increase it in the future. Thus, the EU contribution should be targeted to support the establishment of new forests, assuming that backward processes remain under strict legal restraint. The common agricultural policy (CAP) of the EU should further focus on green direct payment, especially maintaining permanent grassland, which not only supports carbon sequestration but also contributes to the protection of biodiversity (Regulation (EU) No. 1307/2013). In parallel, Lithuania should continue to maintain its permanent grassland [55].

## 5. Conclusions

The prediction accuracy of land use types directly related to GHG accounting and emission/absorption management in the LULUCF sector in Lithuania was above 80% over a five-year period into the future. Land use types whose abundance changed relatively linearly during the last three decades—forest and built-up lands—were predicted with accuracies of 96% and above. The most challenging was the prediction of land use types on agricultural land, i.e., the separation between producing land and grassland. These results were obtained using a compatible methodological approach based on a Markov chain-type model as used by Lithuanian authorities to estimate forest reference levels for the country following the European Union land use, land use change, and forestry regulation for 2021–2030. It should be emphasized that driver variables affecting land

use transformation over time were estimated from information freely available from GIS databases, as the modeling exercise was implemented using standard GIS software.

All scenarios tested suggested that the LULUCF sector in Lithuania would accumulate carbon during the next decade, starting in 2020. The main land use type contributing to the most carbon absorption will remain forest land. Even though the proportion of forest land area in Lithuania is predicted to increase, we did not manage to simulate forest land proportions exceeding 37% of the country's area by either applying land use management approaches as they were applied since 2005 or by introducing additional measures to support forest land expansion. The key factors to improve the GHG balance from the LULUCF sector in the near future, assuming a stable development of forest land and strict deforestation control, are keeping the proportion of producing land and grassland and afforestation of abandoned and uncultivated agricultural lands.

To facilitate $CO_2$ emission/absorption management in the LULUCF sector together with increasing socioeconomic and environmental benefits of Lithuanian rural landscapes, more sophisticated tools to support the monitoring, analysis, and modeling of land-related mitigation activities are needed. Lithuania has developed an original land use monitoring system that is used for GHG reporting, which, up to some level, predetermines land use development projections. However, even though the system is sufficient to fulfil the country's international climate change mitigation commitments, it encompasses a number of limitations in both substantiating the methodology and the way it is operationally implemented. Further research is needed to improve the methodological framework for integrated land management, which can make use of the digital technologies for inventory and decision support to serve the needs of managers and policy makers with a specific focus on GHG management. More specifically, wall-to-wall mapped land use and land use changes would provide better inputs for land use development scenario modeling using the methodological approach tested in this study. The development of spatially explicit land use change scenario modeling and analysis tools could focus on the use of cellular automata and agent-based modeling approaches.

**Author Contributions:** Conceptualization, G.M. and D.J.; methodology, G.M. and D.J.; software, G.M. and D.J.; validation, G.M.; formal analysis, G.M. and D.J.; writing—original draft preparation, G.M.; writing—review and editing, D.J.; visualization, G.M. All authors have read and agreed to the published version of the manuscript.

**Funding:** This research received no external funding.

**Institutional Review Board Statement:** Not applicable.

**Informed Consent Statement:** Not applicable.

**Data Availability Statement:** Data available on request.

**Acknowledgments:** The study was implemented within the framework of a research project following the conditions of the agreement between Aleksandras Stulginskis University and the State Forest Service of the Ministry of Environment of Republic of Lithuania No. 22 from 26 May 2016: "Project development for a forest management and land use scenario modeling subsystem within the National Forest Inventory Information System".

**Conflicts of Interest:** The authors declare no conflict of interest.

## Appendix A

**Table A1.** List of land use subtypes.

| Land Use Subtype | Area Proportion in 2015 * |
|---|---|
| Forest land | 33.78 |
| Arable (producing) land | 32.49 |
| Cultural meadows and pastures | 11.44 |
| Natural grassland | 5.16 |
| Natural grassland covered by trees and brush | 5.06 |
| Cities, settlements and homesteads | 3.84 |
| Natural lakes and rivers | 3.02 |
| Roads and railways | 1.35 |
| Brush | 0.95 |
| Land reclamation ditches | 0.87 |
| Wetlands covered by trees and brush | 0.64 |
| Wetlands | 0.34 |
| Peat bogs | 0.34 |
| Orchards | 0.15 |
| Other built-up land | 0.15 |
| Routes and electricity lines | 0.10 |
| Queries | 0.10 |
| Berry fields | 0.08 |
| Other other land use | 0.07 |
| Other meadows and pastures | 0.02 |
| Other waters and wetlands | 0.02 |
| Short rotation plantations, willow plantations | 0.02 |
| Other producing land | 0.02 |
| Stony land | 0.01 |

* based on the validation data set.

**Table A2.** List of explanatory variables tested to predict the land use transition potential.

| Description of the Variable | Source Database |
|---|---|
| Distance based variables | |
| Distance to the nearest agricultural block in KŽS. If the distance equals 0, then the plot is located in agricultural block | KŽS |
| Distance to the nearest built-up block in KŽS. If the distance equals 0, then the plot is located in built-up block | |
| Distance to the nearest miscellaneous block in KŽS (basically, forest). If the distance equals 0, then the plot is located in miscellaneous block | |
| Distance to the nearest road block in KŽS. If the distance equals 0, then the plot is located on the road | |
| Distance to the nearest block around linear hydrographic object in KŽS. If the distance equals 0, then the plot is located on the linear hydrographic object | |
| Distance to the nearest block around areal hydrographic object in KŽS. If the distance equals 0, then the plot is located on areal hydrographic object | |
| Area proportion-based variables | |
| Proportion of agricultural land in the zone around the NFI sample plot | KŽS |
| Proportion of built-up land in the zone around the NFI sample plot | |
| Proportion of miscellaneous land (basically, forest) in the zone around the NFI sample plot | |
| Proportion of land associated with the road blocks in the zone around the NFI sample plot | |
| Proportion of land associated with the blocks around linear hydrographic object in KŽS in the zone around the NFI sample plot | |
| Proportion of land associated with areal hydrographic object in KŽS in the zone around the NFI sample plot | |
| Proportion of land associated with the miscellaneous blocks with dominance of land not used for agriculture in KŽS in the zone around the NFI sample plot (for the period after 2010 only) | |
| Proportion of protected areas in the zone around the NFI sample plot | SŽNS_DR10LT |
| Proportion of nature heritage areas in the zone around the NFI sample plot | |
| Proportion of protective zones in the zone around the NFI sample plot | |
| Proportion of abandoned agricultural land in the zone around the NFI sample plot | AZ_DRLT |
| Variables available from land declaration data | |
| Proportion of producing land in the zone around the NFI sample plot | Spatial data set on the farmland, cropland and crop types |
| Proportion of berry-field land in the zone around the NFI sample plot | |
| Proportion of orchard land in the zone around the NFI sample plot | |
| Proportion of other producing land in the zone around the NFI sample plot | |
| Proportion of forest land in the zone around the NFI sample plot | |
| Proportion of pastures and meadows in the zone around the NFI sample plot | |
| Proportion of natural grassland in the zone around the NFI sample plot | |
| Proportion of other pastures and meadows in the zone around the NFI sample plot | |
| Proportion of waters and wetlands in the zone around the NFI sample plot | |
| Other variables | |
| Average soil productivity grade in the zone around the NFI sample plot | Dirv_DR10LT |
| Population density in 1 km$^2$ cell, the NFI sample plot belongs to | Population and housing census 2011 |

**Table A3.** Predicted carbon emission and absorption from the LULUCF sector in Lithuania, depending on scenario (numeric values used to build Figure 6, in t $CO_2$ eq./ha).

| Land Use Type | Prediction Years | | | | | |
|---|---|---|---|---|---|---|
| | **2020** | **2025** | **2030** | **2020** | **2025** | **2030** |
| | Reference (2005–2010) | | | Reference (2010–2015) | | |
| Forest land | −1.331 | −1.343 | −1.355 | −1.351 | −1.378 | −1.406 |
| Producing land | 0.519 | 0.535 | 0.546 | 0.460 | 0.468 | 0.428 |
| Grassland | −0.098 | −0.090 | −0.085 | −0.117 | −0.112 | −0.123 |
| Wetland | 0.139 | 0.139 | 0.139 | 0.138 | 0.134 | 0.138 |
| Built-up land | 0.084 | 0.084 | 0.084 | 0.083 | 0.080 | 0.077 |
| Other land | 0.013 | 0.013 | 0.013 | 0.011 | 0.011 | 0.011 |
| GHG balance in LULUCF sector | −0.674 | −0.662 | −0.658 | −0.775 | −0.795 | −0.874 |
| GHG balance in agricultural land | 0.421 | 0.445 | 0.461 | 0.343 | 0.357 | 0.305 |
| | Producing land to forest (2005–2010) | | | Producing land to forest (2010–2015) | | |
| Forest land | −1.345 | −1.372 | −1.392 | −1.458 | −1.419 | −1.449 |
| Producing land | 0.451 | 0.479 | 0.480 | 0.393 | 0.393 | 0.369 |
| Grassland | −0.120 | −0.107 | −0.103 | −0.128 | −0.136 | −0.140 |
| Wetland | 0.139 | 0.139 | 0.139 | 0.138 | 0.138 | 0.138 |
| Built-up land | 0.084 | 0.084 | 0.084 | 0.078 | 0.070 | 0.071 |
| Other land | 0.013 | 0.013 | 0.013 | 0.011 | 0.011 | 0.011 |
| GHG balance in LULUCF sector | −0.778 | −0.764 | −0.780 | −0.966 | −0.943 | −1.001 |
| GHG balance in agricultural land | 0.331 | 0.372 | 0.377 | 0.265 | 0.257 | 0.228 |
| | Grassland to forest (2005–2010) | | | Grassland to forest (2010–2015) | | |
| Forest land | −1.380 | −1.388 | −1.395 | −1.424 | −1.436 | −1.452 |
| Producing land | 0.519 | 0.535 | 0.546 | 0.460 | 0.444 | 0.428 |
| Grassland | −0.091 | −0.084 | −0.080 | −0.112 | −0.116 | −0.119 |
| Wetland | 0.139 | 0.139 | 0.139 | 0.138 | 0.138 | 0.138 |
| Built-up land | 0.084 | 0.084 | 0.084 | 0.067 | 0.069 | 0.071 |
| Other land | 0.013 | 0.013 | 0.013 | 0.011 | 0.011 | 0.011 |
| GHG balance in LULUCF sector | −0.717 | −0.701 | −0.693 | −0.860 | −0.889 | −0.923 |
| GHG balance in agricultural land | 0.427 | 0.451 | 0.466 | 0.348 | 0.329 | 0.309 |
| | No grassland to producing land (2005–2010) | | | No grassland to producing land (2010–2015) | | |
| Forest land | −1.331 | −1.343 | −1.355 | −1.351 | −1.377 | −1.412 |
| Producing land | 0.418 | 0.407 | 0.399 | 0.428 | 0.399 | 0.376 |
| Grassland | −0.134 | −0.136 | −0.137 | −0.128 | −0.136 | −0.141 |
| Wetland | 0.139 | 0.139 | 0.139 | 0.138 | 0.138 | 0.134 |
| Built-up land | 0.084 | 0.084 | 0.084 | 0.083 | 0.080 | 0.077 |
| Other land | 0.013 | 0.013 | 0.013 | 0.011 | 0.011 | 0.011 |
| GHG balance in LULUCF sector | −0.811 | −0.837 | −0.858 | −0.818 | −0.884 | −0.955 |
| GHG balance in agricultural land | 0.285 | 0.271 | 0.261 | 0.300 | 0.263 | 0.235 |

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
