# Peer review of "Modeling Future Land Use Development: A Lithuanian Case"

_land, doi:10.3390/land10040360_

Round 1

Reviewer 1 Report

There is no defined research objective in the abstract 

Lines 457- 60 we can find information "The second type of measure (grassland to forest) was aimed to increase the forest land area on current grassland. It matchesthe first scenario, however, with no conditions regarding the efforts to prevent transformations from producing land into grassland. - Please refer to whether this is not against the policy of the European Union. (e.g. in Poland, you cannot generally afforest grassland, only agricultural land). 

Author Response

Point 1: There is no defined research objective in the abstract 

Response 1: The abstract was modified aiming to specify the research objectives. Also, there were some minor adjustments introduced in the abstract, to avoid overlap with newly inserted text. More specifically, we inserted the following text: ‘This study addresses methodological principles for land use development scenario modeling assumed for use in processes of GHG accounting and management. Associated land use policy implications in Lithuania are also discussed.

Point 2: Lines 457- 60 we can find information "The second type of measure (grassland to forest) was aimed to increase the forest land area on current grassland. It matchesthe first scenario, however, with no conditions regarding the efforts to prevent transformations from producing land into grassland. - Please refer to whether this is not against the policy of the European Union. (e.g. in Poland, you cannot generally afforest grassland, only agricultural land). 

Response 2: The scenarios used in current study were associated both with land use policy options and some sensitivity analyses of different methodological approaches. The last objective could lead also to some deviations from the “most realistic” or “most wanted” futures. The approach regarding grassland-forest land transformations was linked to the processes in Lithuanian land use management and forestry. We should note, that there are some gaps and contradictions in Lithuanian legislation regarding legal protection such important landuses, like agricultural land (arable land, orchards, meadows and pastures) and forest land. E.g. Article 25 of Lithuanian Land Law assumes the opportunity to plant forest on the land assigned for agricultural use, however, the Forest Law backward transformation of forest law into agricultural land is not allowed. Afforestation in agricultural territories usually means forest planting on agricultural land uses, following strict legal procedures. There are several requirements regarding artificial afforestation – land productivity should not be above some margin and afforestation should not damage the land reclamation systems if such are available). However, the Forest Law does now allow to convert naturally grown forest land back into agricultural land after it is inventoried and included into the Forest state cadastre. This is quite common case (that was also assumed in our scenario) that a lot of agricultural land is grown naturally by trees, which can be considered as forest land using, e.g. FAO nomenclature, however, not considered as forest land according to Forest Law. Such land may be treated as abandoned agricultural land with all associated legal consequences and is often cleared from tree vegetation (stopping such activity was mainly assumed in our scenario). However, rather significant share of such land (if the age of naturally regenerated trees gets >20 years) is included into Forest State cadastre and removing trees there would mead illegal forest harvesting. Illegal forest harvesting is considered as more severe violation of legal acts than abandoning of agricultural land (note, that including some land parcel into Forest State Cadastre does not change the official land use type).

Measures associated with increasing the areas of grassland were identified in former lines 476-485.

Reviewer 2 Report

Two small comments:

The Figure 1 (lines 130, 131 etc.) contains actually two relatively independent Figures. The first one is the location of the study area and the second is about the land use changes in the past. I would recommend to divide this Figure into two separated Figures and describe in more details the study area.

There is some confusion with the numbers of tables in the Appendix. The number “Table A2” appears two times – on the line 545 (Table A2. List of explanatory variables …) and on the line 546 (Table A2. Predicted carbon emission …). The last one should be “A3” obviously as on the line 331 in reference to the Appendix 3.

Author Response

Point 1: Two small comments:

The Figure 1 (lines 130, 131 etc.) contains actually two relatively independent Figures. The first one is the location of the study area and the second is about the land use changes in the past. I would recommend to divide this Figure into two separated Figures and describe in more details the study area.

Response 1: Former figure 1 was split into two parts: Figure 1 introducing the ‘Location of the study area’ and Figure 2, displaying the ‘Specification of the study area’. Correspondingly, numbering of figures was changed in further text. Land uses in Lithuania were specified in more details by adding the following text: ‘The area of three land uses important in GHG accounting and management (forest, producing land and grassland) was rather similar (around 28–30%) in 1971’… ‘… (Figure 2). The proportions of forest land and producing land in 2015 were, respectively, 34% and 33%. The proportion of grassland was reduced to 23%, and the proportions of both wetland and built-up land were 5%. Should be noted, that total area of Lithuania is 65,200 km2’. Part of Figure 2 is introduced in subchapter 2.2 on Input data.

Point 2: There is some confusion with the numbers of tables in the Appendix. The number “Table A2” appears two times – on the line 545 (Table A2. List of explanatory variables …) and on the line 546 (Table A2. Predicted carbon emission …). The last one should be “A3” obviously as on the line 331 in reference to the Appendix 3.

Response 2: Sorry for technical issues. We corrected the numbering for the last table (i.e. Table A.3). Correspondingly, we corrected a caption for Figure 6, which mentions the Table A.3.

Reviewer 3 Report

This study was well designed, and the manuscript was very well written. The only suggestion I have is for you to consider sorting Table A1 based on the proportion of area - so that the reader can focus on the main classes. 

Author Response

Point 1: This study was well designed, and the manuscript was very well written. The only suggestion I have is for you to consider sorting Table A1 based on the proportion of area - so that the reader can focus on the main classes. 

Response 2: Thanks for your evaluation. Table A1 was adjusted following your recommendation.

Reviewer 4 Report

Modeling Future Land Use Development: A Lithuanian Case

This article explores methods of informed decision-making to regulate greenhouse gas (GHG) emissions though the effective land use management in Lithuania.  It also forecasts the land use scenarios up to 2030 by utilizing reference data from 2010 to 2015 and other freely available data. It uses almost two and half dozens of variables to see future scenarios of carbon storage (sequestration) if the current land use policy continues. Using the Markov matrices to show the probability of a land use/cover change from forestry to agriculture to pasture and vice-versa from 2020 to 2030, this research predicts carbon balances under various land use and cover scenarios. It argues that limitation of data has created problems in predicting the exact amount of GHG conservation from certain land use and cover, especially, during the transition phases. Overall, the paper has merits to publish because it educates readers about the land use and cover scenarios in Lithuania, and the methodology used in this research is transferrable to similar geographic and socioeconomic and demographic conditions. However, the following issues need to be addressed first.

  1. The English must be polished. There are too many run-on sentences. It takes long time to understand the literal meanings of some of the sentences especially from introduction up to input Data (2.2). The writing is too complex, confusing due to longer and run-on sentences. From introduction to input data section, the whole must be rewritten if the authors wish to publish this article. Other sections also need polishing English.
  2. Please make the research question clearly understandable
  3. What is the suitability of data available from the Lithuanian NFI for modeling land use development?
  4. What is the performance of methodological approaches used to construct forest reference levels in modeling land use development?
  5. What land use developments (do the author mean scenario??) in Lithuania may be expected over the coming decade if current land use policies are continued?

If the authors address these issues and polish the English, the article will be reviewed again. Then the paper will be publishable.

Author Response

Point 1: This article explores methods of informed decision-making to regulate greenhouse gas (GHG) emissions though the effective land use management in Lithuania.  It also forecasts the land use scenarios up to 2030 by utilizing reference data from 2010 to 2015 and other freely available data. It uses almost two and half dozens of variables to see future scenarios of carbon storage (sequestration) if the current land use policy continues. Using the Markov matrices to show the probability of a land use/cover change from forestry to agriculture to pasture and vice-versa from 2020 to 2030, this research predicts carbon balances under various land use and cover scenarios. It argues that limitation of data has created problems in predicting the exact amount of GHG conservation from certain land use and cover, especially, during the transition phases. Overall, the paper has merits to publish because it educates readers about the land use and cover scenarios in Lithuania, and the methodology used in this research is transferrable to similar geographic and socioeconomic and demographic conditions. However, the following issues need to be addressed first.

Response 1: We tried to address all the issues identified. Please, find detailed comments on the modifications done bellow.

Point 2: The English must be polished. There are too many run-on sentences. It takes long time to understand the literal meanings of some of the sentences especially from introduction up to input Data (2.2). The writing is too complex, confusing due to longer and run-on sentences. From introduction to input data section, the whole must be rewritten if the authors wish to publish this article. Other sections also need polishing English.

Response 2: Indeed, the original text did not look good. We accept, that even though the English language was edited by 2 editors from MDPI’s English editing service, the meaning could be improved. We carefully examined the whole text (with specific emphasises on chapters pointed out by the reviewer) trying critically to assess whether the sentence could be made simpler and easier to read. First, we tried the contents to remain the same. Most of the sentences were adjusted in one or another way. Long sentences were split into several part. Some (potentially redundant) text was removed. All the changes made were identified change tracking. Formulation of research questions was changed, hopefully better addressing other points raised by the reviewer.

Point 3: Please make the research question clearly understandable

Response 3: We have to accept that the formulation of research questions was very poor. We changed the whole last paragraph of the Introduction, trying to formulate research questions (even though we did not use such term in the text) which (we hope) better fit the tasks investigated and results achieved. We also suspected that the following points identified by the reviewer (4 to 6) could be due to poor formulation of research questions (see explained below, responses 4 to 6). The part of the text on the research questions was changed to look: ‘First, we ask what is the performance of Markov chain analyses methodological approach in modelling land use development using standard GIS software? To conduct the modelling exercise, we use inputs available from already running in Lithuania inventory projects and freely available geographic databases. Then, we test the capacity of the LULUCF sector in Lithuania to accumulate carbon during the next decade, starting in 2020. For that we project the development of major land use types in Lithuania until 2030 using several land use management scenarios and estimate potential contributions of different land uses on carbon emission/absorption. We hypothesise that the carbon accumulation in the LULUCF sector in Lithuania during the next decade should increase. Finally, we end with a discussion and proposals for both methodological enhancements of modelling solutions and land use management policies’. We also removed some text introducing the chapters of our manuscript.

Point 4: What is the suitability of data available from the Lithuanian NFI for modeling land use development?

Response 4: We hope that this point was raised due to poor initial formulation of research questions. We revised the chapter on research questions (see response 3). Indeed, Lithuanian NFI data on land uses each year starting from 1971 is the only spatially explicit data source, which is fully compatible with carbon accounting in Lithuania. Even though Lithuanian NFI was invented to solve other tasks that land use monitoring, it was delegated such function a decade ago and is responsible for LULUCF data. The discussion includes some thoughts on the suitability of sampling based input data, however, we end with the recommendation for wall-to-wall land use mapping.

Point 5: What is the performance of methodological approaches used to construct forest reference levels in modeling land use development?

Response 5: As above, we revised the paragraph on research questions, and such statement was removed. We accept that that the initial formulation was very unclear. We tried to explain that our focus was on Markov chain analyses (this methodological approach is used also to construct the forest reference levels, but we have to accept, that this is much beyond the scope of current paper). Formally, yes, it is working, the prediction accuracy has been over 80%. We discuss this shortly in the Discussion. We also explain in the Discussion the willingness of Lithuanian authorities to have land use modelling methodologically compatible with forest management scenario modelling.

Point 6: What land use developments (do the author mean scenario??) in Lithuania may be expected over the coming decade if current land use policies are continued?

Response 6: As above – we changed the formulations. We emphasised the focus on carbon conservation trends in the LULUCF sector. As a scenario we assume a narrative of future situations rather than development of modelled phenomenon, land use in our case. Indeed, we do not discuss which development trend should or would occur. Our key findings were that the carbon accumulation in the LULUCF sector in Lithuania during the next decade should increase, with the ‘key factor to improve the GHG balance from the LULUCF sector in the near future’ to be ‘the proportion of producing land and grassland’.

Point 7: If the authors address these issues and polish the English, the article will be reviewed again. Then the paper will be publishable.

Response 7: We carefully checked the whole text and hope that major language issues disappeared.

Round 2

Reviewer 4 Report

Thank you for revising the manuscript.